# Loop-Mediated Isothermal Amplification Assay Using Gold Nanoparticles for Detecting *Treponema pallidum* subspp. *pallidum*

**DOI:** 10.3390/diagnostics14202323

**Published:** 2024-10-18

**Authors:** Saranthum Phurijaruyangkun, Pongbun Tangjitrungrot, Pornpun Jaratsing, Suphitcha Augkarawaritsawong, Sawanya Pongparit, Rungnapa Veeramano, Kularb Tanomnuch, Supatra Areekit, Kosum Chansiri, Somchai Santiwatanakul

**Affiliations:** 1Faculty of Medicine, Srinakharinwirot University, Bangkok 10110, Thailand; saranthum.phurijaruyangkun@g.swu.ac.th; 2Center of Excellence in Biosensors, Panyananthaphikhu Chonprathan Medical Center, Srinakharinwirot University, Nonthaburi 11120, Thailand; pongbun.pao@gmail.com (P.T.); pornpunj@g.swu.ac.th (P.J.); supatraa@g.swu.ac.th (S.A.); 3Faculty of Medical Technology, Rangsit University, Pathum Thani 12000, Thailand; suphitcha.a@rsu.ac.th (S.A.); sawanya.p@rsu.ac.th (S.P.); rungnapa.v@rsu.ac.th (R.V.); 4Clinical and Anatomical Pathology, Panyananthaphikhu Chonprathan Medical Center, Srinakharinwirot University, Nonthaburi 11120, Thailand; nantanomnuch@gmail.com; 5Innovative of Learning, Srinakharinwirot University, Bangkok 10110, Thailand; 6Srinakharinwirot University, Bangkok 10110, Thailand

**Keywords:** *Treponema pallidum*, loop-mediated isothermal amplification (LAMP), gold nanoparticles (AuNPs)

## Abstract

Background: Venereal syphilis in humans is caused by *Trepenoma pallidum* subspp. *pallidum*. A study has shown that 30,302 individuals in Thailand had syphilis in 2020, with a male-to-female ratio of 1:0.8 and the highest incidence rate at ages between fifteen and twenty-four. Methods: This research aimed to develop a loop-mediated isothermal amplification assay using gold nanoparticles (LAMP-AuNPs). Analytical sensitivity, diagnostic specificity, accuracy, and predictive values for each technique are provided. Results: The diagnosis sensitivities of polymerase chain reaction using agarose gel electrophoresis (PCR-AGE), loop-mediated isothermal amplification assay using agarose gel electrophoresis (LAMP-AGE), and LAMP-AuNPs were 116 ng/µL, 11.6 ng/µL, and 11.6 ng/µL, respectively. We evaluated the analytical specificity using PCR and a LAMP-based assay, and there was no cross-reactivity to *Leptospira interrogans*, *Staphylococcus aureus*, *Enterococcus faecalis*, *Escherichia coli*, *Klebsiella pneumoniae*, *Acinetobacter baumannii*, *Pseudomonas aeruginosa*, human immunodeficiency virus (HIV), and healthy humans. After analyzing 400 serum samples of patients suspected of syphilis, the LAMP-AGE and LAMP-AuNPs assays displayed 100% diagnostic sensitivity scores, 91% diagnostic specificity scores, 95.5% accuracy rates, 100% positive predictive values (PPVs), and 91% negative predictive values (NPVs), the positive likelihood ratio (LR+) was 11.11, while the negative likelihood ratio (LR−) was 0. Conversely, for PCR assays displayed 100% diagnostic sensitivity scores, 94.5% diagnostic specificity scores, 97.25% accuracy rates, 100% PPVs, and 94.5% NPVs, LR+ was 18.18, and LR− was 0. Conclusions: The LAMP-AuNPs technique demonstrates rapidity, affordability, and convenience, rendering it well-suited for point-of-care applications in the diagnosis, prevention, and management of pathogenic infections.

## 1. Introduction

*Treponema pallidum* subspp. *pallidum* is a member of the order spirochaetales, family spirochaetaceae, and genus treponema. It causes venereal syphilis in humans, a mostly sexually transmitted disease. Congenital syphilis can also occur when an infected mother transfers her infection to her fetus. Binary logistic regression has been carried out to determine the factors associated with syphilis in pregnant women. The findings showed that, in 2020, the number of syphilis cases among the Thai population reached 30,302 (morbidity rate: 45.2 per 100,000 population). The male-to-female ratio was 1:0.8. The morbidity of syphilis was highest in the 15–24 age group, with a morbidity rate of 144.7 per 100,000 population. In addition, in 2023, 43.6% of the identified syphilis cases were individuals engaged in casual labor or factory work, and 19.1% had co-infections. There were also 3039 reported cases of syphilis in pregnant women. The overall prevalence rate was 0.56%, with a median age of 21.2 years [1,2]. The natural course of untreated syphilis can be separated into three stages: primary, secondary, and latent [3,4,5,6].

In primary syphilis, vasculopathy progression leads to clear endarteritis obliterans in local arteries, resulting in tissue necrosis and, finally, a genital ulcer in the shape of a chancre. In secondary syphilis, the skin lesion along with the various histological patterns and granulomas appearing in the later stage can overlay the aforementioned characteristic features, resolving without treatment. Gumma formation defines tertiary syphilis, including cardiovascular syphilis and psychiatric manifestations [7,8,9,10,11,12,13].

Laboratory diagnosis includes directly detecting *T. pallidum* in clinical material via dark-field microscopy or immunohistochemical examination. Diagnosis is made via both non-treponemal and treponemal antibody serology tests. Non-treponemal tests include a screening test rapid plasma regain (RPR) and Venereal disease research laboratory (VDRL), while treponemal antibodies include confirmatory tests such as fluorescent treponemal antibody absorption (FTA-ABS), *T. pallidum* hemagglutination (TPHA), Treponema pallidum particle agglutination assay (TPPA), Enzyme immunoassay (EIA), and chemiluminescence immunoassay (CIA). Molecular methods such as polymerase chain reaction (PCR) and loop-mediated isothermal amplification (LAMP) methods detect the nucleotide of the pathogen. LAMP is a novel nucleic acid amplification method developed by Notomi et al. in Japan. The LAMP reaction progresses under isothermal conditions through strand displacement, and four different primers are designed to recognize six distinct regions, making it more specific compared to PCR, which recognizes only two distinct regions. The amplification was performed under isothermal conditions between 63 and 65 °C, therefore, was no need to use an expensive thermocycler. This method could potentially replace PCR because of its simplicity, rapidity, and cost-effectiveness [14,15,16]. The LAMP assay is one of the most powerful tools used for gene amplification; with its help, micro-biotic infections can be quickly identified. It can also be rapidly used by various investigators to detect different types of emerging viruses, such as West Nile virus, severe acute respiratory syndrome, dengue virus, and Japanese encephalitis virus [17,18].

Thus, in this study, we describe the development of a molecular technique combining a LAMP assay with gold nanoparticles (LAMP-AuNPs) for diagnosing *T. pallidum* subspp. *pallidum* and an ultrasensitive assay for *T. pallidum* based on amplifying the gene encoding the high-conversion pathogen-specific 47 kDa membrane immunogen (*Tpp47*) [19,20,21,22,23,24,25,26]. LAMP-AuNPs is widely used to detect pathogens. Gold nanoparticles (AuNPs) interact with 5′-SH-C6-labeled DNA probes (thiol) through disulfide bonds (S-S), and surface plasmon resonance (SPR) of AuNPs shows a strong and broad peak around 520–540 nm. The detection process is carried out sequentially: (1) the functionalization of *TPP47* gene-specific ssDNA probes onto AuNPs, (2) the hybridization between these ssDNA probes and the syphilis LAMP products, and (3) the modulation of the stability of the DNA functionalized AuNPs by increasing the ionic strength of the solution with MgSO_4_. As MgSO_4_ concentrations increase, the DNA functionalized AuNPs in blank samples, and test samples containing only primers or the target DNA without sequence complementary to ssDNA probe (negative samples) become unstable. This instability arises due to salt’s screening effect, leading to immediate aggregation, which is evidenced by a color change from red to blue, purple, or gray, as well as a shift in surface plasmon resonance (SPR) toward longer wavelengths (SPR at 600–650 nm). In contrast, when Syphilis LAMP products with sequence complementarity to the ssDNA probe are present (positive samples), the DNA functionalized AuNPs remain stable and dispersed (SPR at 520–525 nm) (Figure 1). This stability is attributed to increased electrostatic repulsion and steric hindrance resulting from hybridization, preventing any observable colorimetric change or shift in the surface plasmon peak. Typically, the aggregation of free AuNPs is induced by agents such as NaCl, MgCl_2_, and MgSO_4_. This color change can be observed visually or with UV–visible spectral analysis. The LAMP-AuNPs assay is fast, cost-effective, and convenient, making it ideal for point-of-care pathogen detection, prevention, and control [27,28,29,30].

## 2. Materials and Methods

### 2.1. Sample Collection

In total, 400 serum samples were consecutively collected from patients suspected of having syphilis at Panyananthaphikkhu Chonprathan Medical Center, Srinakharinwirot University, between May 2022 to May 2023. In the laboratory, the specimens underwent routine screening using RPR and TPHA tests. Diagnostic criteria for syphilis were based on the guidelines outlined in the Bangrak Sexual Transmitted Infections (STIs) Center’s National Guideline for Syphilis Testing [31]. As part of the study’s protocol, serum samples from healthy blood donors were collected to serve as negative controls. The approval for this study was granted by the Institutional Ethics Committee of Srinakharinwirot University (SWUEC/X-099/2565) in April 2022.

### 2.2. DNA Extraction

The DNA was extracted from 200 μL of serum samples from suspected syphilis patients and healthy humans, as well as from colony suspension of non-*Treponemal pallidum* strains including *Leptospira interrogans*, *Staphylococcus aureus*, *Enterococcus faecalis*, *Escherichia coli*, *Klebsiella pneumoniae*, *Acinetobacter baumannii*, and *Pseudomonas aeruginosa*. The extractions were done using QIAamp DNA kits from Qiagen^®^, Hamburg, Germany, according to the manufacturer’s instructions. The extracted DNA samples were stored at −20°C until required for testing.

### 2.3. Primers and DNA Probe Design

PCR and LAMP primers were designed to amplify the *Tpp47* gene sequence of *T. pallidum* subspp. *pallidum* using genetic data from GenBank (Bethesda, MD, USA), accession number AE000520.1:622266-623570. The primers were designed using Primer Explorer V5 (https://primerexplorer.jp/e/ (accessed on 10 March 2022)) and Primer3web V4.1.0 (https://primer3.ut.ee/ (accessed on 10 March 2022)). The LAMP primer set includes outer primers (F3 and B3) and inner primers (FIP and BIP), targeting six specific sequences on the DNA (patent submission number: 2103001639). For the LAMP-AuNPs assay, DNA probes were labeled with thiol (SH-C6) at the 5′ end. Primer and probe analysis confirmed their identity as *T. pallidum* subspp. *pallidum* using NCBI nucleotide BLAST with 100% similarity. These primers and probes were purchased from Pacific Science Co. Ltd. (Biobasic, Markham, ON, Canada).

### 2.4. Optimization of PCR Reaction Conditions

PCR amplification was carried out in a 25 μL reaction mixture containing sterile water, 10× PCR buffer (Vivantis, Carlsbad, CA, USA), 0.4 μM of F3 and B3 primers, 2.8 mM of MgCl_2_ (Vivantis, USA), 0.2 mM of dNTPs (New England Biolabs, Ipswich, MA, USA), 2 units of Taq DNA polymerase (Vivantis, USA), and 1 μL of DNA template. Milli-Q water was served as a blank negative control. To determine the optimal annealing temperature, the PCR reaction was conducted using the T100 Thermal Cycler (Bio-Rad, Hercules, CA, USA). The cycling conditions included an initial denaturation step at 95 °C for 5 min, followed by 30 cycles of denaturation at 95 °C for 30 s, annealing at 50–60 °C for 30 s, and extension at 72 °C for 30 s. PCR products were analyzed via 2.0% agarose gel electrophoresis (AGE) at 100 volts for 30 min, stained with ViSafe Red Gel Stain (Vivantis, Darul Ehsan, Malaysia), and visualized on a UV transilluminator.

### 2.5. Optimization of LAMP Reaction Conditions

The LAMP reaction was conducted in a 25 μL reaction mixture containing sterile water, 1× supplied buffer (New England Biolabs, USA), 0.2 μM each of FIP and BIP primers, 2.0 μM each of F3 and B3 primers, 1.6 mM dNTPs (New England Biolabs, USA), 0.5 M betaine (Sigma-Aldrich, USA), varying concentrations of MgSO_4_ (3.5–6.5 mM; New England Biolabs, USA), 8 units of *Bst* 2.0 DNA polymerase (large fragment; New England Biolabs, USA), and a DNA template. Milli-Q water was used as a negative control to check for cross-contamination. The reaction mixture was incubated at temperatures ranging from 60 to 65 °C for different durations (15, 30, 45, and 60 min) in a heat block. LAMP products were analyzed via 2.0% AGE at 100 volts for 30 min, stained with ViSafe Red Gel Stain (Vivantis, Darul Ehsan, Malaysia), and visualized using a UV transilluminator (Analytik Jena US, St Upland, CA, USA).

### 2.6. Preparation of the AuNPs-Labeled DNA Thiol Probe

To prepare AuNPs with a thiol-labeled DNA probe as an anchor, following a documented protocol. Initially, 4 mL of 10 nM colloidal AuNPs was mixed with 20 μL of 100 μM thiol-labeled DNA probe in a 15 mL centrifuge tube. The tube was covered with aluminum foil and placed in a shaker at 55 °C at 100 rpm for 24 h. Simultaneously, a washing buffer (10 mM phosphate buffer pH 7.0, 0.01% sodium dodecyl sulfate (SDS) or surfactant solution, and 100 mM NaCl or salt buffer) was prepared and transferred to a mixing tube. Throughout this process, the solution’s color remained red, indicating no AuNPs’ aggregation. The mixing tube was also wrapped in aluminum foil and shaken at 45 °C at 100 rpm for 48 h.

After that, the solution containing the AuNPs-labeled DNA thiol probe was transferred to a clean 1.5 mL microcentrifuge tube and then centrifuged at 13,000 rpm at 4 °C for 25 min. The supernatants were carefully removed, and the sediment was washed twice with 500 μL of washing buffer (10 mM Phosphate-buffered saline; PBS), pH 7.4, 150 mM NaCl, and 0.1% SDS). Subsequently, the AuNPs-labeled DNA thiol probe was washed three times using 100–200 μL of washing buffer to remove excess thiol-labeled DNA probe. The DNA thiol probe labeled with AuNPs was detected using a UV-Vis spectrophotometer (NanoDrop 2000, Thermo Fisher Scientific, Waltham, MA, USA). The optical properties of the DNA thiol probe labeled with AuNPs demonstrated SPR spectra ranging from 515 to 540 nm. The solution of the AuNPs-labeled DNA thiol probe was stored at 4 °C (not frozen) until used.

### 2.7. Optimization of LAMP-AuNPs Reaction Conditions

After the LAMP assay was conducted, the AuNPs-labeled DNA thiol probe was added to LAMP products to hybridize at 63 °C for 10 min, using various concentrations of MgSO_4_ ranging from 1 to 10 mM. The ratio of AuNPs-labeled DNA thiol probe to LAMP product to MgSO_4_ was maintained at 1:1:1. The hybridization product could be observed via the naked eye, as the positive samples retained a red color with no AuNPs-labeled DNA thiol probe aggregation, while negative samples turned from red to blue, purple, or gray owing to AuNPs aggregation induced by MgSO_4_. The solution color was confirmed using UV-Vis spectrophotometer analysis. The SPR spectra of positive samples appeared at 500–525 nm, while those of negative samples appeared at 600–650 nm.

### 2.8. Analytical Sensitivity and Specificity Tests

The analytical sensitivity (limit of detection) of the PCR- and LAMP-based assays was measured using ten-fold serial dilutions of the positive control, genomic DNA of *T. pallidum* DNA (ATCC^®^ BAA-2642SD^TM^, Manassas, VA, USA), from 11.6 ng/µL to 1.16 fg/µL. The analytical specificity of all assays was tested by measuring the cross-reactivity with the extracted DNA of *L. interrogans*, *S. aureus*, *E. faecalis*, *E. coli*, *K. pneumoniae*, *A. baumannii*, *P. aeruginosa*, and HIV, as well as a healthy human sample, and milli-Q water as negative controls.

### 2.9. The Diagnostic Testing Accuracy

Four hundred serum samples, including those reactive with RPR titer ≥1:16 and nonreactive to RPR, were tested and confirmed using the TPHA serology test. The diagnostic sensitivity, specificity, accuracy, positive predictive value (PPV), and negative predictive value (NPV) were calculated based on laboratory testing and physician assessment, whether the patient had the disease or not, using 2 × 2 cross-tabulated diagnostic tests.

Diagnostic sensitivity measures the percentage of true positives (TPs) relative to all positive cases (TPs + false negatives (FNs)). Diagnostic specificity measures the percentage of true negatives (TNs) relative to all negative cases (false positives (FPs) + TNs). Accuracy is the proportion of correctly identified cases of TPs plus TNs relative to the total cases. The PPV represents the percentage of TPs relative to all positive test results (TPs + FPs), and the NPV indicates the percentage of TNs, which relative to all negative test results (FNs + TNs) (Table 1). Additionally, likelihood ratios are useful for interpreting diagnostic test results in everyday clinical practice. The likelihood ratio of a positive test result (LR+) is sensitivity divided by (1− specificity), and the likelihood ratio of a negative test result (LR−) is (1− sensitivity) divided by specificity. Note that in the equations, the likelihood ratio requires converting percentages of sensitivity and specificity into decimal form.

## 3. Results and Discussion

### 3.1. Optimization of PCR Amplification

The annealing temperature for PCR amplification varied from 50 to 60 °C at 2 °C, and the intervals were compared to determine the optimal annealing temperature. The appropriate annealing temperature that yielded the most distinct PCR amplicon band of 192 bp without the presence of nonspecific bands was identified at 58 °C (Figure 2).

### 3.2. Optimization of LAMP Amplification

#### 3.2.1. The Concentration of MgSO_4_

Various concentrations of MgSO_4_ one of the critical factors was achieved in the LAMP reaction. To optimize the concentration of MgSO_4_ for LAMP amplification, a range of 3.5–6.5 mM were evaluated, and the optimal results were obtained at concentrations of 4.5 mM (Figure 3).

#### 3.2.2. Temperature Variations

The optimal amplification temperature for LAMP was determined in a gradient from 60 to 65 °C. The amplification patterns with ladder band clarity of the LAMP assay at 63 °C demonstrated the high amplification efficiency, which was the optimal amplification temperature (Figure 4).

#### 3.2.3. Reaction Time Variations

The optimal reaction time for LAMP amplification was determined at 63 °C for 15, 30, 45, and 60 min. The LAMP amplicons were successfully generated at all time points, and the amplification products were most distinctly detected at 60 min. Consequently, a reaction time of 60 min was selected as the optimal duration for this LAMP assay (Figure 5).

### 3.3. The Optimal Conditions for LAMP-AuNPs

Different concentrations of MgSO_4_ were tested in the range of 1–10 mM for the LAMP-AuNPs assay. The MgSO_4_ concentration of 4–10 mM was able to distinguish between positive results appearing in red and negative results in blue to purple, which were visible to the naked eye. In this study, a MgSO_4_ concentration of 5 mM was used, which showed the most obvious color change of the reaction (Figure 6). The SPR peak for the AuNPs-labeled DNA thiol probe was found at a wavelength of 525 nm, the positive LAMP-AuNPs peak was found at a wavelength of 525 nm, and the negative LAMP-AuNPs peak was found at a wavelength of 625 nm (Figure 7).

### 3.4. Analytical Sensitivity and Specificity Tests

The analytical sensitivity of PCR, LAMP, and LAMP-AuNPs was determined using a 10-fold serial dilution of *T. pallidum* DNA (ATCC^®^ BAA-2642SD^TM^) from 11.6 ng/µL to 1.16 pg/µL. The limits of detection of the PCR, LAMP, and LAMP-AuNPs assays were 116 pg/µL, 11.6 pg/µL, and 11.6 pg/µL, respectively (Figure 8a–c).

The specificity of the PCR, LAMP, and LAMP-AuNPs assays was determined using *Leptospira interrogans*, *Staphylococcus aureus*, *Enterococcus faecalis*, *Escherichia coli*, *Klebsiella pneumoniae*, *Acinetobacter baumannii*, *Pseudomonas aeruginosa*, Human Immunodeficiency Virus, and healthy human sample. All assays showed no cross-reactivity with all other non-Treponema pallidum strains (Figure 9a–c).

### 3.5. The Diagnostic Testing Accuracy

The diagnostic testing accuracy: diagnostic sensitivity, diagnostic specificity, positive predictive values (PPV), negative predictive values (NPV), accuracy, positive likelihood ratio (LR+), and negative likelihood ratio (LR−) for PCR-AGE, LAMP-AGE, and LAMP-AuNPs using data from 400 serum samples (Table 2).

## 4. Discussion

Currently, *T. pallidum* subspp. *pallidum* can be detected using conventional and molecular-based assays, but the newly approved gold-standard method is the antibody test. Although cost-effective, the antibody test has limitations due to its low sensitivity. In this study, we developed the LAMP assay combined with the AuNPs method to detect *T. pallidum* using specific primers and DNA probes based on the *Tpp47* gene. This approach offers a rapid screening test for diagnosing *T. pallidum* subspp. *pallidum* infections, with the process completed in approximately 60 min.

The LAMP-AuNPs assay is useful as a diagnostic method for syphilis, which is not time-consuming and does not require expensive equipment such as a thermocycler. LAMP-AuNPs assay reacted specifically with *T. pallidum* subspp. *pallidum* and no cross-reactivity with other organisms was found in the clinical specimen, such as *L. interrogans*, *S. aureus*, *E. faecalis*, *E. coli*, *K. pneumoniae*, *A. baumannii*, *P. aeruginosa*, and HIV. Furthermore, a comparison of the sensitivity of LAMP-AuNPs and PCR was performed for the detection of syphilis in clinical specimens. The result showed that the LAMP-AuNPs assay was better than PCR. In addition, the diagnostic sensitivity of both LAMP-based assays was 100%, indicating a very high sensitivity. Additionally, the diagnostic specificity was 91%, accuracy was 95.5%, the PPV was 100%, and the NPV was 91%, LR+ was 11.11, LR− was 0. The likelihood ratio for a positive result (LR+) quantifies the extent to which the odds of the disease increase following a positive test result, and the likelihood ratio for a negative result (LR−) quantifies the degree to which the odd of the disease decrease when a negative test result. Likelihood ratios > 1 show an association with disease, whereas ratios < 1 show an association with the absence of disease. The results in this study demonstrated that the LR+ of PCR and LAMP-AuNPs was shown > 1, indicating a large and often conclusive increase in the likelihood of disease, while the LR− of PCR and LAMP-AuNPs was <0.1, indicating a large and often conclusive decrease in the likelihood of disease.

We analyzed the UV-Vis spectra of the AuNPs after they were functionalized with ssDNA probes (Figure 6). Compared with citrate-capped AuNPs, the surface plasmon band of the DNA-functionalized AuNPs shifted slightly from 520 to 525 nm. This shift indicates a change in the refractive index around the AuNPs, confirming the successful attachment of the ssDNA probe to the AuNPs surface. A similar effect was observed in previous studies with 13–20 nm AuNPs [27].

The optimal MgSO_4_ concentration was 4–10 mM, ensuring that repulsive forces were completely screened. This concentration is lower than previously reported. Further tests with MgSO_4_ concentrations ranging from 10 to 100 mM showed a clear colorimetric change and a surface plasmon spectral shift at 30 mM [28]. The experimental principle of signal detection for visualizing LAMP products via the post-hybridization of biotin-labeled DNA probes with streptavidin-conjugated AuNPs was also tested with MgSO_4_ concentrations ranging from 10 mM to 1 M. Precipitates formed when the salt concentration increased to 250 mM [29]. Additionally, an allele-specific LAMP assay with gold nanoparticles was used to screen single-nucleotide polymorphisms (SNPs) as disease biomarkers. In this case, 20 mM of MgCl_2_ was added, and the colorimetric change was characterized by UV–Visible spectroscopy using a microplate reader [30].

In our study, we compared the results of our LAMP-AuNPs with other previous studies in Table 3. The other studies used different types of specimens, such as swabs from skin lesions, blood, or CSF, to detect the *Tpp47* gene. However, our study detected the *Tpp47* gene from the serum of patients used for serological tests without the need for collecting additional specimens [32]. Our LAMP-AuNPs showed 100% sensitivity compared to the physician’s confirmed diagnosis based on clinical symptoms and laboratory results. The specificity of our study was up to 91%. The 9% false positives may be due to the assay’s ability to detect the *Tpp47* gene before the patient developed clinical symptoms or specific and non-specific antibodies. However, further medical follow-up and additional laboratory tests are required to confirm whether the test results are false positives or an early detection of syphilis.

## 5. Conclusions

The diagnosis of syphilis by the serological method involves two steps to confirm the infection. The first step is to detect non-specific antibodies, and the second is to detect specific antibodies. To detect both antibodies requires at least two weeks for the antibodies to develop against *Treponema pallidum* subspp. *pallidum* infection. As a result, the serological methods cannot detect the early stages of syphilis, which may lead to more severe in different stages of the disease. However, the molecular biological test directly detects genes to the syphilis pathogen and thus can detect an early stage of infection, when the infection has spread to the lymph nodes and bloodstream. Compared with serological tests, the LAMP-AuNPs test is simple, quick, highly sensitive, specific, and straightforward and does not require expensive equipment or specialized equipment like thermal cyclers. It has the potential to significantly aid in managing patients with syphilis, particularly in regions with a high prevalence and limited resources. Enhancing LAMP diagnostics by integrating a single-step approach with a core solution will improve its suitability as a point-of-care diagnostic test.

This study is part of the development of a test kit for infectious disease diagnosis using various techniques based on the LAMP principle by modifying the detection method of LAMP products. The study found that LAMP-AuNPs can accurately diagnose patients with and without syphilis, and this technique is currently being evaluated for patenting and further development into a commercial test kit. In the future, LAMP-AuNPs assay may be of great importance in the management of syphilis patients, especially in areas with high prevalence but limited resources. The LAMP-AuNPs technique is rapid and convenient, making it ideal for the development of a point-of-care diagnostic test for the management of syphilis and other infectious diseases.

## Figures and Tables

**Figure 1 diagnostics-14-02323-f001:**
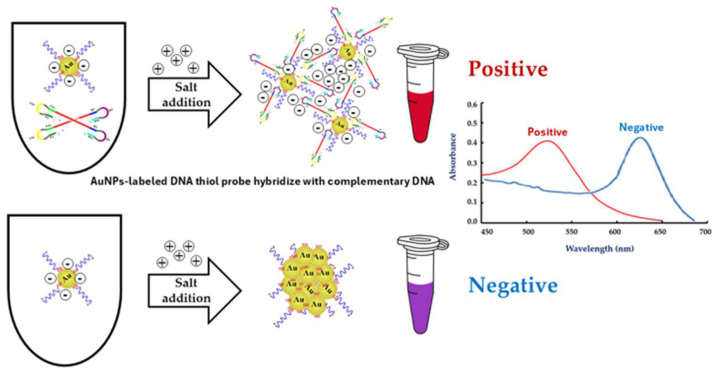
A schematic illustration of LAMP-AuNPs assay.

**Figure 2 diagnostics-14-02323-f002:**
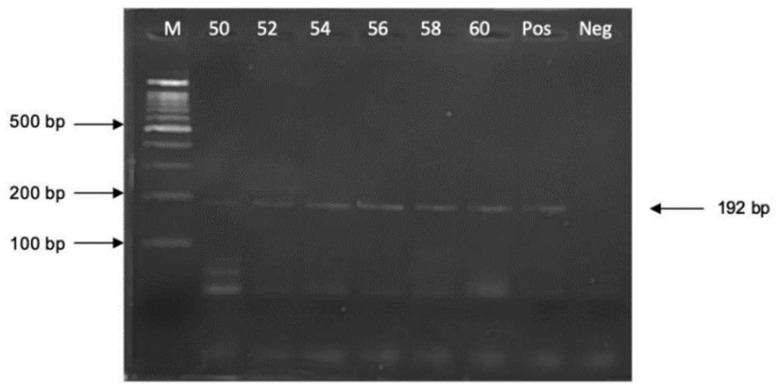
Lane “M” represents DNA ladder marker 100 bp plus (Vivantis, USA); “Pos” represents positive control; and “Neg” represents negative control.

**Figure 3 diagnostics-14-02323-f003:**
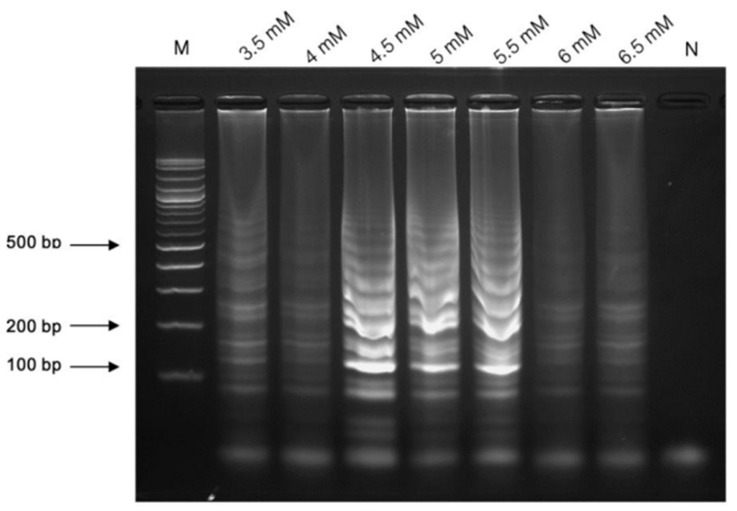
The optimization of LAMP assay in the range of 3.5–6.5 mM. Lane “M” represents a 100 bp plus DNA ladder marker of Vivantis, USA, and “Neg” is the negative control.

**Figure 4 diagnostics-14-02323-f004:**
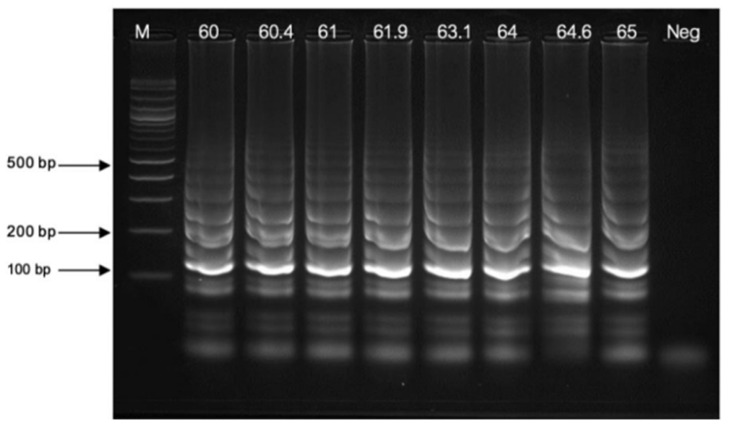
The optimization of thermal conditions ranging between 60 and 65 ℃. Lane “M” represents a 100 bp plus DNA ladder marker of Vivantis, USA, and “Neg” is the negative control.

**Figure 5 diagnostics-14-02323-f005:**
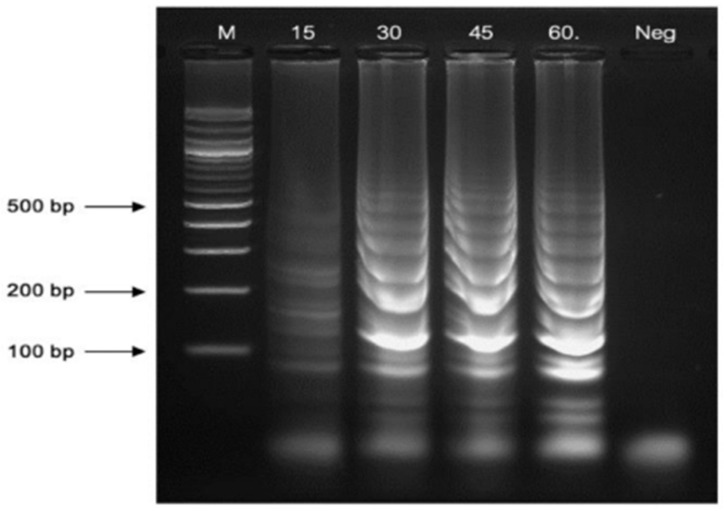
The optimal reaction durations were observed at 15- and 60-min. Lane “M” represents a 100 bp plus DNA ladder marker of Vivantis, USA, and “Neg” is the negative control.

**Figure 6 diagnostics-14-02323-f006:**
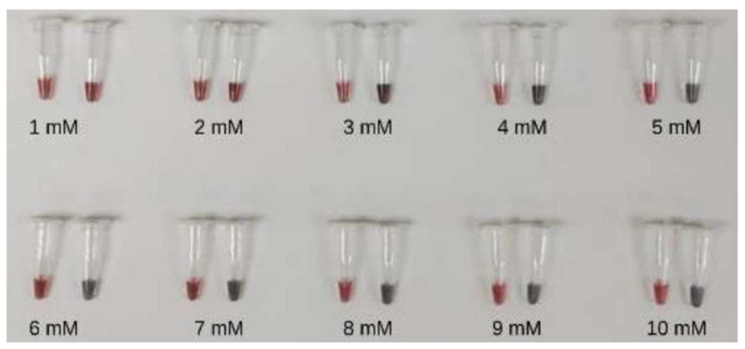
Variation in MgSO_4_ concentrations with 1–10 mM LAMP-AuNPs.

**Figure 7 diagnostics-14-02323-f007:**
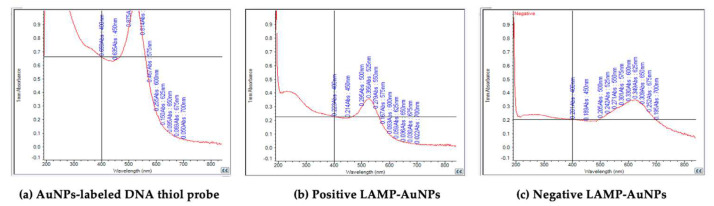
The SPR spectra: (**a**) the AuNPs−labeled DNA thiol probe; (**b**) the positive LAMP−AuNPs; (**c**) the negative LAMP−AuNPs.

**Figure 8 diagnostics-14-02323-f008:**
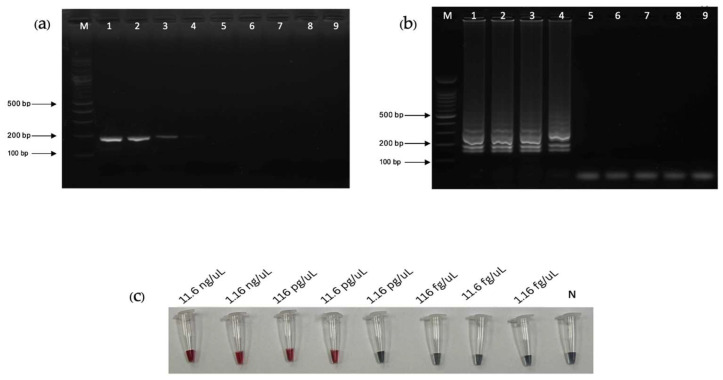
Lane “M” represents DNA ladder marker 100 bp plus (Vivantis, USA). DNA 10-fold dilutions: 1: 11.6 ng/µL; 2: 1.16 ng/µL; 3: 116 pg/µL; 4: 11.6 pg/µL; 5: 1.16 pg/µL; 6: 116 fg/µL; 7: 11.6 fg/µL; 8: 1.16 fg/µL; and 9: negative control. (**a**) PCR with agarose gel electrophoresis (PCR-AGE). (**b**) LAMP with agarose gel electrophoresis (LAMP-AGE). (**c**) LAMP combined with gold nanoparticles (LAMP-AuNPs).

**Figure 9 diagnostics-14-02323-f009:**
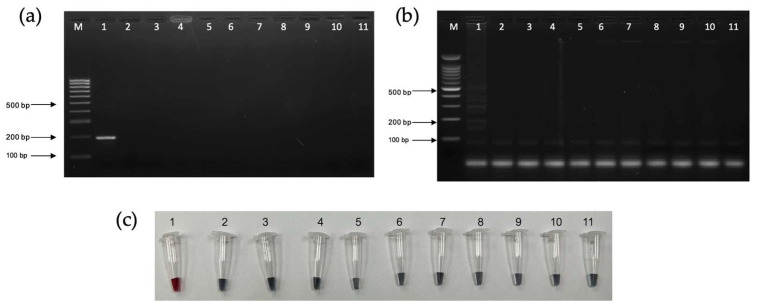
Lane “M” represents DNA ladder marker 100 bp plus (Vivantis, USA). 1. Trepenoma pallidum subspp. pallidum; 2. Leptospira interrogans; 3. Staphylococcus aureus; 4. Enterococcus faecalis; 5. Escherichia coli; 6. Klebsiella pneumoniae; 7. Acinetobacter baumannii; 8. Pseudomonas aeruginosa; 9. Human Immunodeficiency Virus (HIV); 10. healthy human; and 11. negative control. (**a**) PCR with agarose gel electrophoresis (PCR-AGE). (**b**) LAMP with agarose gel electrophoresis (LAMP-AGE). (**c**) LAMP combined with gold nanoparticles (LAMP-AuNPs).

**Table 1 diagnostics-14-02323-t001:** A 2 × 2 cross-tabulation of the diagnostic tests; disease presence means that the RPR is reactive at ≥1:16, and the TPHA is reactive.

Diagnostic Test	Gold Standard		
	Disease	No Disease	
**Positive**	True positive (TP)	False positive (FP)	PPV = TP/(TP + FP)
**Negative**	False negative (FN)	True negative (TN)	NPV = TN/(FN + TN)
	Sensitivity = TP/(TP + FN)	Specificity = TN/(FP + TN)	

**Table 2 diagnostics-14-02323-t002:** The diagnostic testing accuracy of PCR- and LAMP-based techniques based on data from syphilis-unknown samples.

The Diagnostic Testing Accuracy	PCR-AGE	LAMP-AGE	LAMP-AuNPs
Diagnostic sensitivity	100%	100%	100%
Diagnostic specificity	94.5%	91%	91%
Accuracy	97.25%	95.5%	95.5%
Positive predictive value (PPV)	100%	100%	100%
Negative predictive value (NPV)	94.5%	91%	91%
Positive likelihood ratio (LR+)	18.18	11.11	11.11
Negative likelihood ratio (LR−)	0	0	0

**Table 3 diagnostics-14-02323-t003:** Comparison of the specimen, the sensitivity, and the specificity results of the other studies using different Nucleic Acid Amplification Tests (NAATs) for syphilis diagnosis [33,34,35,36,37,38,39,40,41].

Methods	Sample Size	Specimens	Target	Sensitivity %	Specificity %	Reference
Routine PCR	288	Swab	*Tpp47*	89.1	99.1	33
Multiplex real-time PCR	15	Swab	*Tpp47* *Taqman*-LNA	100	100	34
Real-time PCR	99	Ulcer lesion	*Tpp47*	100	97.14	35
Nested PCR	329	Swab	*Tpp47*	92.0	95.0	36
Nested PCR	315	Blood	*Tpp47*	90.3	100	37
Real-time PCR	122	CSF	*Tpp47*	58.0	67.0	38
Routine PCR	124	CSF	*Tpp47*	75.8	86.8	39
Nested PCR	40	CSF	*Tpp47*	42.5	97.0	40
Nested PCR	262	Whole blood	*Tpp47*	53.6	-	41
LAMP-AuNPs	400	Serum	*Tpp47*	100	91.0	This study

## Data Availability

The data presented in this study are available on request from the corresponding author.

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
