# Peer review of "Loop-Mediated Isothermal Amplification Assay Using Gold Nanoparticles for Detecting Treponema pallidum subspp. pallidum"

_diagnostics, 2024, doi:10.3390/diagnostics14202323_

Round 1
Reviewer 1 Report
Comments and Suggestions for Authors
Why did the authors choose LAMP? While there are a lot of new isothermal POCTs that work at 37-42'C with less primer than LAMP.
The likelihood ratio (LR) for used methods is suggested to be calculated to compare their detection potential (correlation between disease with/without infection). According to the manuscript's data, LR+ of LAMP is better than the PCR technique.
Author Response
Comments 1: [Why did the authors choose LAMP? While there are a lot of new isothermal POCTs that work at 37-42'C with less primer than LAMP.] |
Response 1: Thank you for pointing this out. We agree with this comment. Therefore, we have added more advantages of LAMP assay. The LAMP reaction progresses under isothermal conditions through strand displacement, and four different primers are designed to recognize six distinct regions, making it more specific compared to PCR which recognizes only two distinct regions. The amplification was performed under isothermal conditions between 63 and 65 °C, therefore was no need to use an expensive thermocycler. (line 69-73) LAMP assay is simple, quick, highly sensitive, specific, and straightforward and does not require expensive equipment or specialized equipment like thermal cyclers. (line 346-348) Furthermore, this study is part of the development of a test kit for infectious disease diagnosis using various LAMP-based techniques. The method for detecting LAMP products has been modified and the technique is under evaluation for patenting and further development into a commercial test kit. Therefore, the use of other isothermal techniques such as Recombinase Polymerase Amplification (RPA) is subject to patent restrictions and must be avoided.
|
Comments 2: [The likelihood ratio (LR) for used methods is suggested to be calculated to compare their detection potential (correlation between disease with/without infection). According to the manuscript's data, LR+ of LAMP is better than the PCR technique.] |
Response 2: Thank you for pointing this out. We strongly agree with this comment. Therefore, we have added the likelihood ratio as you suggested because the likelihood ratios are another way to show the efficiency of a diagnostic test, in addition to sensitivity, specificity, and predictive value. In our study, the results of likelihood ratio help indicate that the LAMP-AuNPs assay can be used as syphilis diagnostic tests, since they can show the discrimination between the patient with disease and non-disease. (on page 5 and 9) |
Comments 3: [Your suggestion for improving the methods description, the results and conclusion |
Response 3: We revised the manuscript according to your comments. The revision of the method, result and conclusion are highlighted in yellow. |

Reviewer 2 Report
Comments and Suggestions for Authors
Dear Author,
The presented MS ´Loop-mediated isothermal amplification assay using gold nanoparticles for detecting Treponema pallidum subsp. pallidum´ is very intresting and presented very well. The application of LAMP technique for detection is certainly has edge for developing the POC diagnostic system.
However i have some suggestions,
1- A schematic diagram for introductin would also help the young readers and the people has less background in this technology.
2- References are not following the Journal format.
3- A comparision table against other technology and advancement of presented methodology is added and it would be good to add in a comparison discussion about the improvements and the challenges of the technology methods.
4- The conclusion is written very short; please add some more points and also the commercial aspects towards POC development.
5- The experimental methodology is eaplained well however little parapharasing in the section 2 will help for better understanding.
Author Response
Comments 1: [A schematic diagram for introduction would also help the young readers and the people has less background in this technology.] |
Response 1: Thank you for pointing this out. We agree with this comment. Therefore, we have added schematic diagram of LAMP-AuNPs assay in figure 1, as show on page 3
|
Comments 2: [References are not following the Journal format.] |
Response 2: Thank you for pointing this out. We agree with this comment. Therefore, we have changed the reference format according to instructions for authors, to use the numbered style guide |
Comments 3: [A comparision table against other technology and advancement of presented methodology is added and it would be good to add in a comparison discussion about the improvements and the challenges of the technology methods.] |
Response 3: Thank you for your comment. Accordingly, we have revised our conclusions by adding the Comparison of the specimen, the sensitivity, and the specificity results of the other studies using different Nucleic Acid Amplification Tests (NAATs) for syphilis diagnosis was shown in table 3 on page 11 Comments 4: [The conclusion is written very short; please add some more points and also the commercial aspects towards POC development.] Response 4: Thank you for your suggestion. Accordingly, we have revised our conclusions considering your comments. Such revisions include the evaluation of the LAMP-AuNPs method for patentability and the development of it into a commercial test kit. This test kit would be used as a point-of-care diagnostic test for managing syphilis, as well as other infectious diseases. (on page 11)
Comments 5: [The experimental methodology is explained well however little paraphrasing in the section 2 will help for better understanding.] Response 5: Thank you a lot for your suggestion. We have revised the manuscript according to your comments. The revision of the methods is highlighted in yellow. |
